# Epidemiological observations and management challenges in extrapedal mycetoma: A three-decade review of 420 cases

**Abubakr Abdalla Mohammed Alhaj, Eiman Siddig Ahmed, Abeer Hassan, Ahmed Hassan Fahal** *

The Mycetoma Research Center, University of Khartoum, Khartoum, Sudan

* ahfahal@mycetoma.edu.sd, ahfahal@hotmail.com

## Abstract

Mycetoma is a serious, destructive, disfiguring chronic granulomatous inflammatory disease affecting the subcutaneous tissues that spread to involve the skin, deep tissues and bone. The disease predominately affects the limbs, and extrapedal mycetoma is rarely reported. The reported extrapedal ones are characterised by high morbidity and mortality. This communication reports on 420 patients with extrapedal mycetoma seen and managed at the Mycetoma Research Centre (MRC), University of Khartoum, between January 1991 and December 2021. In this descriptive, cross-sectional, hospital-based study, the electronic records of all mycetoma-confirmed patients seen during the study period were carefully and meticulously reviewed. The confirmed patients with extrapedal mycetoma were included in this study. The study included 420 patients with extrapedal mycetoma, 298 (70.7%) had eumycetoma, and 122 (29.3%) had actinomycetoma. There were 343 male patients (81.7%) and 77 (18.3%) females, with a male-to-female ratio of 4:1. Their ages ranged between 1.5 and 95 years, with a median of 28 years. Most of the patients were students and farmers. The majority of patients were from El Gezira, North Kordofan, and the White Nile States. Mycetoma was painful in 21%, and a family history of mycetoma was recorded in 11.5% of patients. The buttocks (37.9%) and head and neck (16.9%) were affected most. Less frequently affected sites were the trunk and back (12%) each, abdominal and chest walls (4.5%) each and loin (1%). The prominent clinical presentation findings were multiple sinuses discharging grains (55%), massive swellings (46%), and lymphadenopathy (11.5%). Less commonly observed clinical findings were local hyperhidrosis (5.3%) and dilated tortuous veins close to mycetoma lesions (0.5%). The study showed that 204 patients (48.6%) had clinical improvement in terms of decreased lesion size and healing of sinuses following medical therapy. Sixty-six patients (15.7%) had no noticeable improvement. The lesion continued progressing despite treatment in 44 patients (10.5%). In the study, 118 patients were on regular follow-up, and in this group, a cure was documented in 25 patients (21.1%) with eumycetoma and 23 (19.4%) with actinomycetoma. Post-operative recurrence among eumycetoma patients was 40%, with a 1% mortality rate. The treatment outcome was unsatisfactory, characterised by a low cure rate, high recurrence (40%) and

**Funding:** The author(s) received no specific funding for this work.

**Competing interests:** The authors have declared that no competing interests exist.

follow-up dropout (57%) rates. This emphasises the importance of early case detection and management, objective health education programmes and thorough patient counselling to urge people to seek treatment early and reduce dropouts.

## Author summary

This communication reports on 420 patients with extrapedal mycetoma seen at the Mycetoma Research Centre, Khartoum, Sudan, from January 1991 to December 2021. Males were predominantly affected. Their ages ranged between 1.5 and 95 years; most were students and farmers. The majority of patients were from El Gezira, North Kordofan and White Nile States. Family history of mycetoma was recorded in only a few. The buttock was affected most, followed by the head, neck, back, chest, and abdominal wall. The prominent clinical presentation findings were massive lesions with multiple sinuses discharging grains and lymphadenopathy. Eumycetoma was the most encountered type. The treatment outcome was unsatisfactory, characterised by a low cure rate and high recurrence and follow-up dropout rates. This emphasises the importance of early case detection and management, health education programmes, and patient counselling to encourage them to seek treatment early and reduce poor treatment outcomes and dropouts.

## Introduction

Mycetoma, a forgotten, neglected disease, is prevalent in tropical and subtropical areas but has been documented globally [1,2]. It is a deep implantation infection due to subcutaneous inoculation of the causative microorganisms, normally inhabited in the soil and environment in the endemic areas, via trivial trauma, notably thorn pricks [3,4,5]. Mycetoma is classified as actinomycetoma or eumycetoma according to its causative microorganisms. Actinomycetoma is caused by filamentous bacteria (actinomycetes), and the frequent ones are *Nocardia asteroides*, *Nocardia brasiliensis*, *Streptomyces somaliensis*, and *Actinomadura madurae* [6,7,8]. The eumycetoma is a mycetoma type caused by different fungi, and the most common are *Madurella mycetomatis* and *Falciformispora spp.* [9,10,11].

Mycetoma is seen in all age groups, but it usually affects young adults between 20 and 40 years old, and 64% of patients are under 30 years old at presentation [12]. Children account for 30% of the reported cases [13]. Males are predominantly affected by mycetoma, with a male-to-female ratio of 3.7:1 [1,2,14]. In most reported series, farmers, manual workers, and students are most often affected [2,15].

Although eumycetoma and actinomycetoma are caused by diverse causative microorganisms, yet, they present with almost similar clinical manifestations, and the cause is an enigma [1,2]. The disease initially manifests as a painless subcutaneous mass, gradually increasing in size, then multiple sinuses and discharge of purulent or seropurulent discharge containing grains appear [1,2,16–18].

Mycetoma was reported in different body sites but was more frequent in the exposed parts subjected to local trauma [3,19]. Thus, the foot (76%) and hand (8%) were reported most [1,2]. However, in endemic regions, no part is immune. Extrapedal mycetoma is not a common disease presentation [20–25]. Its demographic features and clinical presentation are unknown, and the management outcome is unclear. Still, there is great difficulty in establishing the diagnosis of mycetoma, and the situation becomes more difficult when it affects extrapedal sites [26,27]. The risk factors for extrapedal mycetoma are unknown, which is essential in designing

a prevention or control programme. Although extrapedal mycetoma confers more morbidity and mortality, it has not been given sufficient attention. Furthermore, there is no recent comprehensive report on extrapedal mycetoma in Sudan, which has the highest disease burden globally. With this background, this study set out to determine the demographic characteristics features of affected patients, the infection risk factors, if any and patients' management outcomes.

## Patients and methods

### Ethics statement

The study's ethical clearance was obtained from the Soba University Hospital Institutional Review Board, University of Khartoum, No. (SUH/IRB/2022/34).

This descriptive, cross-sectional hospital-based study was conducted at the Mycetoma Research Centre (MRC), University of Khartoum, Khartoum, Sudan. The study included 420 patients with confirmed extrapedal mycetoma seen in the period from January 1991 to December 2021. In this study, extrapedal mycetoma is defined as mycetoma in any body part except the upper and lower limbs.

The diagnosis of mycetoma was confirmed by careful interview, meticulous clinical examinations, and specific investigations, including fine needle aspiration for cytology (FNAC), histopathological examination of surgical biopsies, grains culture and molecular techniques to identify the microorganism to the species level. Various imaging techniques were used, including lesional ultrasound examination, magnetic resonance imaging (MRI), and computed tomography (CT). No consents were obtained as these tests were part of the routine workup of patients at the MRC.

The data were collected from the patients' electronic medical records into a specially designed data collection sheet for analysis. All patients were anonymised.

### Data analysis

The data were cleaned, coded, and managed by the Statistical Package for Social Sciences software (SPSS Statistics 27). Appropriate descriptive statistical tests were used, and that included the measures of frequency (frequency, per cent), measures of central tendency (mean, median and mode), and measures of dispersion or variation (variance, SD, standard error, range, and coefficient of variation). The results were summarised as percentages for categorical variables, mean ± standard error of the mean (SEM), and median for continuous variables.

## Results

### The studied patients' demographic characteristics

There were 420 patients with confirmed extrapedal mycetoma, constituting 4.3% of the total mycetoma patients seen at the MRC during the study period. There were 343 male patients (81.7%) and 77 (18.3%) females. Male to female ratio is 4.4:1. The patients' ages ranged between 1.5 and 95 years, with a median of 28 years (31.1 ± 0.7 standard error). Most patients at presentation were young adults in their third and fourth decade of life, 142 (33.8%) and 105 (25%), respectively, followed by adolescents (17.9%), Table 1.

By occupation, 88 patients (20.9%) were students, and 86 (20.5%) were farmers. Only ten patients (2.38%) were shepherds. 10.47% of this study's patients were housewives, Table 1.

The study showed that 128 patients (30.4%) were from El Gezira State, 51 patients (12.1%) were from North Kordofan State, and 48 patients (11.4%) were from White Nile State. Six patients (1.42%) were from neighbouring countries, South Sudan and Chad, with three

**Table 1. The study population demographic characteristics.**

|  |  | No. | % |
|---|---|---|---|
| **Sex** | Male | 343 | 81.60% |
|  | Female | 77 | 18.40% |
| **Age group** | 01–09 | 05 | 1.20% |
|  | 10–19 | 75 | 17.90% |
|  | 20–29 | 142 | 33.80% |
|  | 30–39 | 105 | 25.00% |
|  | 40–49 | 39 | 9.30% |
|  | 50–59 | 29 | 6.90% |
|  | 60–69 | 17 | 4.00% |
|  | > = 70 | 08 | 1.90% |
| **Occupation** | Student | 88 | 20.9% |
|  | Farmer | 86 | 20.47% |
|  | Worker | 60 | 14.2% |
|  | Housewife | 44 | 10.47% |
|  | Unemployed | 22 | 5.23% |
|  | Driver | 19 | 4.52% |
|  | Shepherd | 10 | 2.38% |
|  | Seller | 05 | 1.19% |
|  | Other | 74 | 17.61% |
|  | Missing | 12 | 2.85% |

patients each. (Fig 1). Most patients with eumycetoma were from El Gezira (27%), White Nile (9.6%), and Sennar (8.8%) states, Fig 1.

## Clinical presentation

The disease duration at presentation ranged between one month and 50 years. In 270 patients (64.28%), its duration ranged between one and four years. Only 77 (18.3%) had the disease for less than one year, Table 2.

The majority of patients, 275 (65.47%), had grain discharge, and that was black in 207 patients (75.2%), yellow in 37 (13.45%), white in 22 (8%), and red in four (1.45%). Sixty-one patients (14.5%) had a history of a sinus discharge, 33 (7.85%) had pus, and 16 (3.8%) had blood discharge, Table 2.

One hundred and two patients (24.2%) had recalled a history of local trauma at the mycetoma, and 68 patients were unsure (16.1%). Different types of traumas were mentioned, including thorns, pricks, stones, cuts, injections, falls and football game trauma. Most of the traumas were considered minor. Mycetoma lesion was painful in 90 patients (21%), and 29 (6.9%) had pain occasionally, Table 2.

Concurrent medical conditions, including diabetes mellitus, hypertension, Hepatitis B (HBV), tuberculosis, renal transplant, and skin dermatitis, were recorded in 5.6% of patients. A family history of mycetoma was present in 11.5%. Ten patients gave a history of mycetoma in a first-degree relative, while seven patients had second-degree relatives. There is no significant statistical difference between the two mycetoma types regarding the family history, $p > 0.01$.

In this study, 285 patients (67.85%) had previous mycetoma surgical excisions done elsewhere prior to presentation to MRC. The operations numbers ranged from one (44%) to ten (0.2%). The surgery was performed under general, spinal, and local anaesthesia, Table 2.

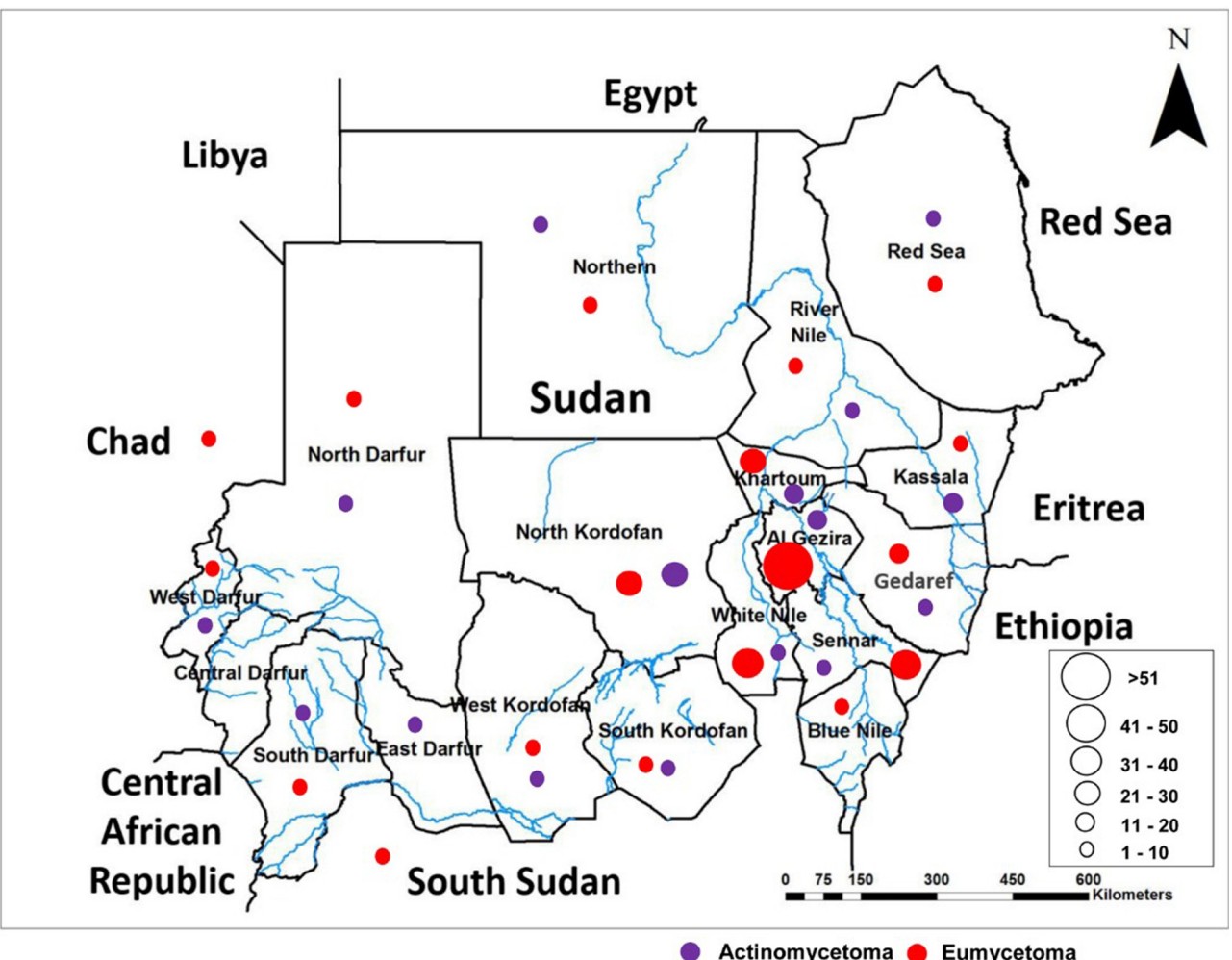

**Fig 1. Distribution of extrapedal eumycetoma and actinomycetoma patients across the different states as seen at the Mycetoma Research Centre from 1991–2021, Sudan.** Created by ArcGIS, version 3. https://www.arcgis.com/home/item.html?id=7ae5637c804b4b6e80abb85d0ca2c26c.

The obtained data showed that 54.8% of them had recurrent disease, and only 36 patients (8.8%) had previous medical treatment for mycetoma elsewhere, Table 2.

There were 169 patients (40.23%) with buttock mycetoma, and the bilateral sides were affected in 15 (3.57%). Spread to the perianal region and perineum were documented in 14 patients (3.33%). There was also an associated inguinal lesion in eight patients with massive gluteal eumycetoma, Fig 2.

The head was the second most common site for extrapedal mycetoma seen in 71 patients (16.9%). The occiput and the forehead were the most affected head regions. The infection was confined to the scalp in 22 patients, but at the occiput, it was deeply spreading.

The back was affected in 52 patients (12.38%), mostly in the lower back 24 (5.71%). The perineum and perianal area were involved in 18 patients (4.3%). Both chest and abdominal walls were equally affected, 4.5% each. The lower anterior abdominal wall was the commonest affected area in the abdominal wall Fig 2

Less commonly affected sites were the ear seen in two young males and ended with devastating complications, including hearing loss, facial nerve palsy, and ultimately death in one patient. Eye mycetoma was documented in eight patients. Two patients with eye

**Table 2. The clinical presentation of the study population.**

|  |  | No. | % |
|---|---|---|---|
| **Disease Duration** | <1 | 77 | 18.3% |
|  | 1–4 | 117 | 27.8% |
|  | 5–9 | 102 | 24.2% |
|  | 10–14 | 51 | 12.1% |
|  | 15–20 | 27 | 6.4% |
|  | 20–25 | 26 | 6.1% |
|  | >25 | 17 | 04% |
|  | Missing | 03 | 0.71% |
| **Course** | Sudden | 01 | 0.23% |
|  | Gradual | 216 | 51.4% |
|  | Progressive | 176 | 41.9% |
|  | Missing | 27 | 6.42% |
| **Discharged grains colour** | Black | 207 | 49.2% |
|  | White | 22 | 5.2% |
|  | Yellow | 37 | 8.8% |
|  | Red | 04 | 0.95% |
|  | None | 71 | 16.9% |
|  | Unidentified | 05 | 01.1% |
|  | Missing | 74 | 17.6% |
| **Sinus discharge** |  |  |  |
| **Pus** | Yes | 33 | 7.85% |
|  | No | 387 | 92.1% |
| **Blood** | Yes | 16 | 03.8% |
|  | No | 404 | 96.2% |
| **Purulent** | Yes | 04 | 0.95% |
|  | No | 416 | 99.04% |
| **Seropurulent** | Yes | 08 | 01.9% |
|  | No | 412 | 98.1% |
| **No discharge** | Yes | 46 | 11% |
|  | No | 374 | 89% |
| **Pain** | Yes | 90 | 21% |
|  | No | 288 | 68.5% |
|  | Sometime | 29 | 06.9% |
|  | Missing | 13 | 03.09% |
| **Trauma history** | Yes | 102 | 25.00% |
|  | No | 238 | 58.30% |
|  | Not Sure | 68 | 16.70% |
| **Medical problem** | No | 356 | 94.40% |
|  | Yes | 21 | 5.60% |
| **Previous surgery & number** | No | 125 | 29.8% |
|  | Yes | 285 | 67.9% |
|  | 1–2 | 239 | 56.9% |
|  | 3–4 | 29 | 6.9% |
|  | 5–6 | 05 | 03% |
|  | 7–8 | 03 | 0.1% |
|  | 10 | 01 | 0.2% |

(*Continued*)

**Table 2.** (Continued)

|  |  | No. | % |
|---|---|---|---|
| **Anaesthesia** | General | 125 | 48.10% |
|  | Local | 59 | 22.70% |
|  | Spinal | 40 | 15.40% |
|  | Combined types | 36 | 13.80% |
| **Previous treatment** | No | 374 | 91.20% |
|  | Yes | 36 | 8.80% |
| **Recurrence** | Yes | 219 | 54.80% |
|  | No | 181 | 45.30% |
| **Family history** | Yes | 38 | 11.50% |
|  | No | 292 | 88.50% |

actinomycetoma had eye destruction, and one had eye enucleation. Few patients had myce-toma in the loin, groin and genitalia, Fig 2.

In this study, 15 patients (3.57%) had multiple trunk or head lesions with associated upper or lower limb lesions. There was no evidence of direct extension from the limbs. The foot was the most affected site associated with other extra pedal mycetoma.

This study revealed that 185 patients (44.04%) had a massive mycetoma lesion (more than 10 cm in diameter) at presentation, only 45 patients (10.71%) had a small lesion (less than 5 cm in diameter), and 125 patients (29.76%) had moderate lesions (5 and 10 cm in diameter). Forty-eight (11.90%) patients presented postoperatively, where surgery was done elsewhere with no clinically apparent lesion at presentation. In some patients, the lesions were detected by ultrasound examination only, Table 3.

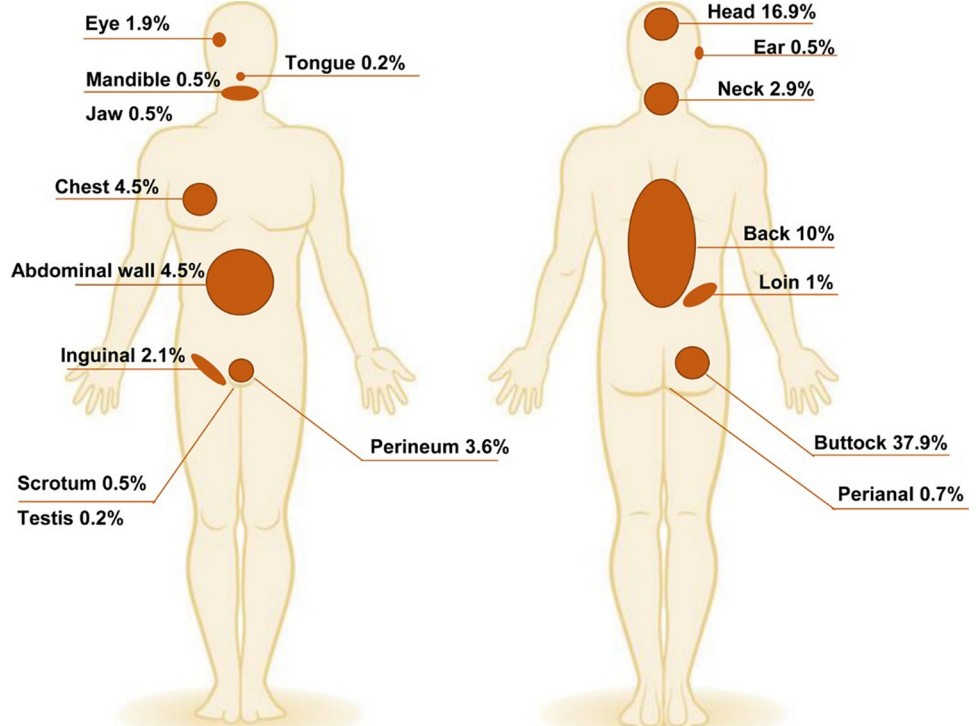

**Fig 2. The anatomical distribution of mycetoma lesions among the study population.**

**Table 3. The clinical examination findings among the study population.**

|  |  | No. | % |
|---|---|---|---|
| **Lesion Size** | Small | 45 | 10.71 |
|  | Medium | 125 | 29.76 |
|  | Massive | 185 | 44.04 |
|  | No mass detected (Post-operative) | 48 | 11.42 |
|  | Missing | 17 | 04.04 |
| **Consistency** | Soft | 30 | 07.14 |
|  | Firm | 260 | 61.90 |
|  | Hard | 08 | 01.90 |
|  | Cystic | 03 | 0.71 |
|  | Missing | 121 | 28.80 |
| **Sinuses** | No | 98 | 23.33 |
|  | Yes | 306 | 72.85 |
|  | Missing | 16 | 3.80 |
| **Number of sinuses** | One | 21 | 05.00 |
|  | Multiple | 131 | 31.19 |
|  | Missing | 268 | 63.80 |
| **Sinuses activity** | Active | 59 | 14.04 |
|  | Healed | 93 | 22.14 |
|  | Non-active | 12 | 02.85 |
|  | Active & healed | 81 | 19.28 |
|  | Active & non-active | 04 | 0.95 |
|  | Non-active–healed | 08 | 1.90 |
|  | All | 04 | 0.95 |
|  | Missing | 261 | 62.1 |
| **Grains** | None | 190 | 54.40 |
|  | Black | 137 | 39.30 |
|  | White | 07 | 2.00 |
|  | Yellow | 11 | 3.20 |
|  | Red | 01 | 0.30 |
|  | Unidentified | 03 | 0.90 |
|  | Missing | 71 | 16.90 |
| **Discharge** | Pus | 56 | 13.33 |
|  | Blood | 07 | 1.66 |
|  | Purulent | 04 | 0.95 |
|  | Seropurulent | 23 | 5.47 |
|  | No discharge | 210 | 50 |
|  | Missing | 39 | 9.28 |
| **Lymph nodes** | No | 301 | 71.66 |
|  | Yes | 39 | 9.28 |
|  | Missing | 80 | 19.04 |
| **Varicose veins** | No | 314 | 74.76 |
|  | Yes | 02 | 0.47 |
|  | Missing | 104 | 24.76 |
| **Local sweating** | Yes | 17 | 0.40 |
|  | No | 303 | 72.14 |
|  | Missing | 100 | 23.80 |

Lesion sinuses were seen in 306 patients (72.85%). They were multiple in 31.19% and single in 05.00%. At presentation, 156 patients (37.1%) had sinuses discharging grains, healed sinuses seen in 93 patients (22.14%), and non-active sinuses in 12 patients (03.85%), Table 3 and Figs 3 and 4.

Regional lymphadenopathy was detected in 39 patients (9.28%). Most of the affected lymph node groups were inguinal from the gluteal, abdominal wall, genitalia and back lesions, cervical groups from head mycetoma and axillary groups from chest wall mycetoma. Dilated tortuous veins proximal to the mycetoma lesion were observed in two patients (0.47%) with

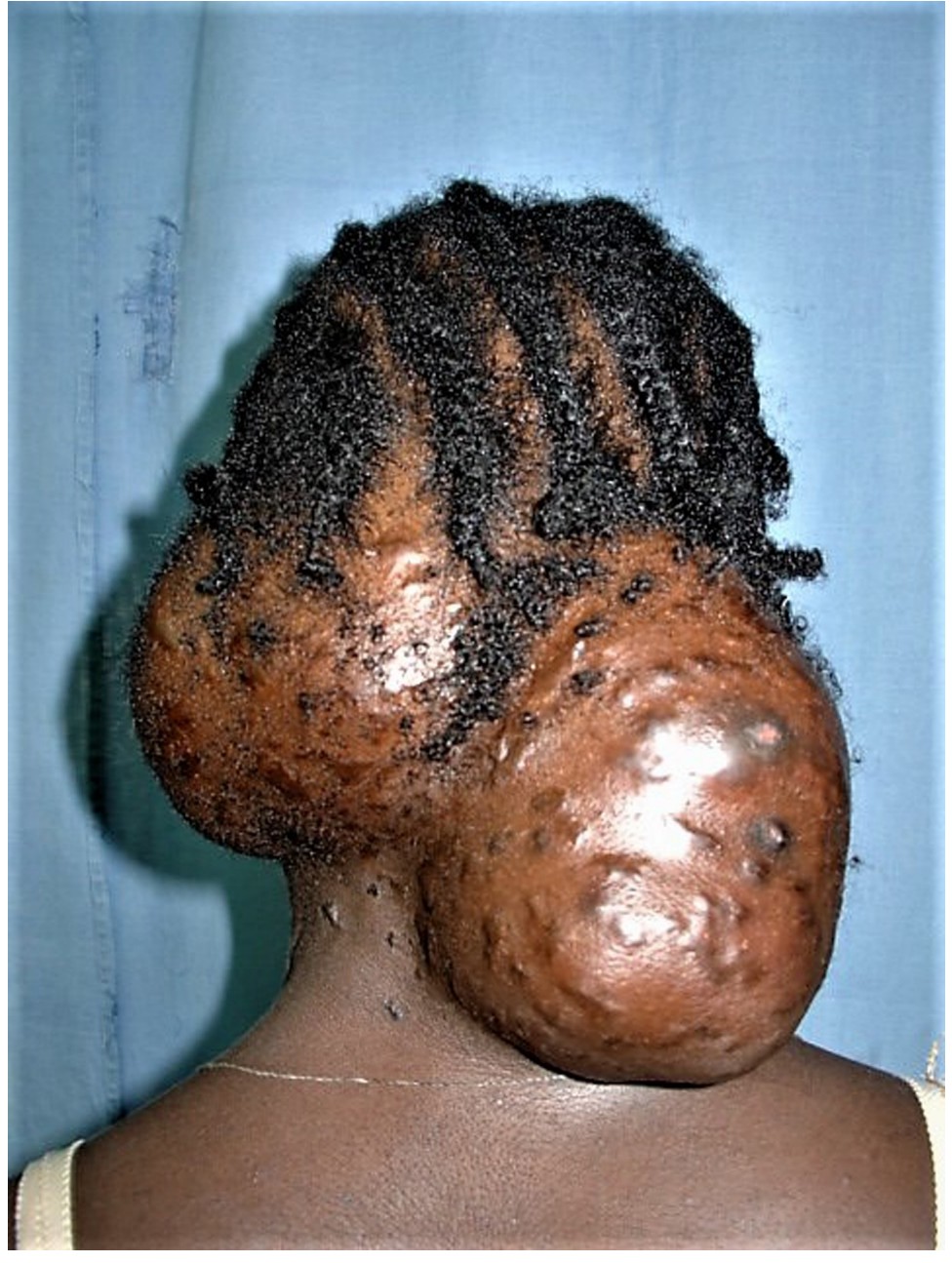

**Fig 3. Showing massive head and neck actinomycetoma.**

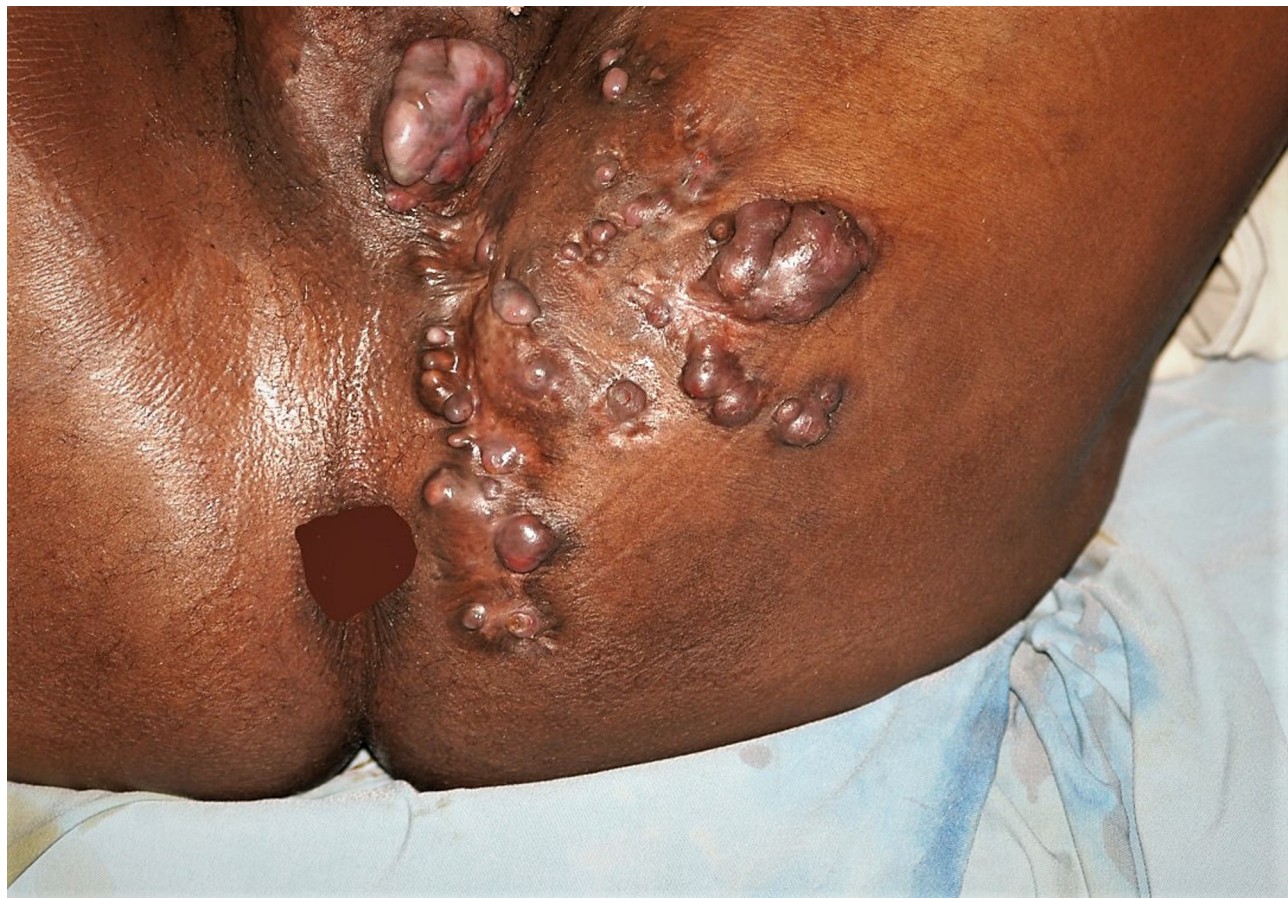

**Fig 4. Showing perineal eumycetoma extending locally and to the scrotum.**

massive and medium-sized lesions; one had bilateral gluteal actinomycetoma, and the other had lower back eumycetoma. Local hyperhidrosis was noted in 17 patients (0.40%).

## Diagnosis

Various diagnostic tools were used to establish the diagnosis of mycetoma among the study group. This was done according to the MRC management guidelines [28]. However, for various reasons, not all the investigations were done for every patient. In this study, 133 patients (31.7%) had an X-ray examination, which was normal in 68 (16.1%). The detected radiological changes are due to the disease local spread to the bones, and that included soft tissue masses seen in 52 patients (39.1%), periosteal reactions in 20 patients (15.0%), and bone destruction in 13 patients (9.8%). Ultrasound examination of mycetoma lesion was done for 194 patients (46.6%) and showed evidence of eumycetoma in 76.7% and actinomycetoma in 15.5%. MRI and CT were done for 69 and 41 patients, respectively, to determine the deep tissue invasion.

Fine needle aspiration and cytology (FNAC) was performed in 203 patients (48.30%) and yielded evidence of *Madurella mycetomatis* in 155 patients (36.9%). In this study, 193 had surgical biopsies and histopathological examinations, which identified causative organisms in 161 patients (82.8%). Of these, two rare microorganisms were identified: *Scedosporium boydii* and *Aspergillus species*, and they are pale-grain eumycetoma causative organisms and the diagnosis was further confirmed by culture and sequencing techniques. The histopathological findings

were nonspecific in 14 patients (7.3%), and that included tissue granuloma or inflammatory process with no identifiable grains. In ten patients, grain cultures revealed *M. mycetomatis* (n = 8) and *Streptomyces somaliensis* (n = 2).

The final diagnosis in this series was eumycetoma, confirmed in 297 patients (70.7%), and the common organism was *M. mycetomatis*, detected in 237 patients (79.79%). Actinomycetoma was diagnosed in 122 patients (29.3%), and the common causative organisms were *Streptomyces somaliensis* detected in 50 patients (16.7%), *Actinomadura madurae* in seven patients (2.3%), and *Actinomadura pelletieri* in six patients (2%). Rare cases included one patient with maxillary sinus had aspergilloma, and another one had back eumycetoma due to *Scedosporium boydii* infection.

## Complications

Several complications were detected among the studied patients. Secondary bacterial infection was seen in 18 patients. *Staphylococcus aureus* and Proteus species were the most common isolated organisms, which was more common among eumycetoma patients (94.4%). The disease was progressive in a patient with the middle ear and mastoid cavity eumycetoma, not responding to treatment and ended in hearing loss and facial nerve paralysis. A patient with a massive vulval actinomycetoma caused by *A. pelletieri* had urethral stricture, bladder outlet obstruction, and acute urine retention. Two patients with back eumycetoma that progressed into paraplegia, and one of them died. Other complications included eye loss, blindness, and deafness in a massive progressive cranial actinomycetoma. The eye was surgically removed in one patient with an advanced eye eumycetoma. Spastic quadriplegia had complicated neck mycetoma in one patient. Two patients had faecal incontinence due to gluteal eumycetoma.

## Mycetoma treatment

Mycetoma treatment depends on the causative microorganism, disease type, site, and extension. Regarding eumycetoma, various antifungal agents combined with wide local excision and repetitive tissue debridement were used. In the past, the antifungals used were griseofulvin and ketoconazole; both have been stopped. Currently, oral itraconazole is used at 400 mg/day.

During this study, actinomycetoma was treated by numerous antibiotic combinations, including streptomycin sulphate and dapsone, streptomycin and trimethoprim-sulfamethoxazole and rifampicin. However, most recently, Amoxicillin/clavulanic acid and trimethoprim-sulfamethoxazole were prescribed because of their strong safety profile. Amikacin and trimethoprim-sulfamethoxazole combination is used for severe cases.

## Follow-up and management outcomes

Only 343 patients (81.6%) were followed at the Mycetoma Clinic; 118 (28%) had regular follow-ups, and 77(18.4%) had no follow-ups after their initial presentations. At present, only 49 patients continue follow-up care (11.7%).

The study showed that 204 patients (48.6%) had clinical improvement in terms of decreased lesion size and healing of sinuses following medical therapy. Sixty-six patients (15.7%) had no noticeable improvement. The lesion continued progressing despite treatment in 44 patients (10.5%).

In the study, 118 patients were on regular follow-up, and a cure was documented in 25 patients (21.1%) with eumycetoma and 23 (19.4%) with actinomycetoma. Post-operative recurrence among eumycetoma patients was 40%, with a 1% mortality rate. Table 4.

**Table 4. The patients' treatment outcomes distribution according to the mycetoma type among patients with regular follow-up.**

| Site | Diagnosis | | | | |
|---|---|---|---|---|---|
| | Eumycetoma | | Actinomycetoma | | Total |
| | Cured | Not cured | Cured | Not Cured | |
| Abdominal Wall | 0 | 1 | 1 | 2 | 4 |
| Back | 4 | 6 | 3 | 2 | 15 |
| Back & Gluteal | 0 | 1 | 0 | 0 | 1 |
| Chest | 1 | 2 | 1 | 0 | 4 |
| Ear | 0 | 1 | 0 | 0 | 1 |
| Eye | 0 | 0 | 2 | 0 | 2 |
| Eye lid | 1 | 0 | 0 | 0 | 1 |
| Gluteal | 15 | 22 | 2 | 3 | 42 |
| Gluteal, Perineum & Scrotal | 0 | 1 | 0 | 0 | 1 |
| Gluteal, inguinal | 1 | 0 | 0 | 0 | 1 |
| Head | 2 | 7 | 10 | 7 | 26 |
| Neck | 0 | 0 | 1 | 2 | 2 |
| Inguinal | 1 | 0 | 1 | 1 | 3 |
| Multiple | 0 | 3 | 0 | 1 | 4 |
| Jew | 0 | 0 | 1 | 0 | 1 |
| Perineum | 0 | 3 | 1 | 2 | 5 |
| Scrotum | 0 | 1 | 0 | 0 | 1 |
| Testis | 0 | 1 | 0 | 0 | 1 |
| Total | 25 | 50 | 23 | 20 | 118 |

## Surgical treatment, cure, and post-operative recurrence

Eighty-five patients had surgical excisions, 73 (85%) had one surgical excision, 70 patients had eumycetoma, 24 (34.2%) were cured, and three patients had actinomycetoma, and one was cured. Twelve patients (15%) had multiple surgical excisions due to recurrence. Ten patients (11.7%) had two surgical operations, and two had three surgeries. Seven patients in this group were cured.

Four patients underwent colostomy to divert the faecal matter from the perianal mycetoma lesions, and that included transverse and permanent colostomy. Of these, one had actinomycetoma, while the other three had eumycetoma.

This study documented postoperative recurrence in 34 patients (40%). Furthermore, one actinomycetoma patient had developed recurrence on two occasions after stopping medical treatment.

Seven patients developed postoperative wound infection; all had eumycetoma, and the commonly isolated organisms were *Staphylococcus aureus*, *Pseudomonas aeruginosa* and Proteus species.

## Mortality among the studied population

In this study, four (1.16%) died during the follow-up period. Three of them were male patients, and they had mycetoma in the head in two patients, one in the ear and one in the back. Of the two patients with head lesions, one had actinomycetoma, and one had eumycetoma. The eumycetoma in the ear was due to *M. mycetomatis*, while the back eumycetoma was due to *Scedosporium boydii*.

## Discussion

The medical literature indicates that extra-pedal mycetoma poses a significant medical challenge, characterised by relentless disease progression, high morbidity, severe complications,

and a risk of mortality [20–25]. Most reported extrapedal cases presented with prolonged disease duration before seeking medical attention. Hence, patients presented with advanced disease and noticeable disfigurement [20–25]. Treatment outcomes are consistently suboptimal, characterised by high mortality and morbidity [20–25]. Based on these serious findings, this study set out to study the extrapedal mycetoma patients treated at the MRC over the past three decades, constituting the largest reported series.

This study reported on 420 patients with confirmed extrapedal mycetoma, constituting 4.3% of the total mycetoma patients seen at the MRC during the study period, which is less frequent than that of the previously reported series [17, 29–31]. A report from the main mycological diagnostic centres in Mexico on 3,933 mycetoma patients, mainly actinomycetoma (96.5%) seen in 54 years, revealed that extrapedal mycetoma accounted for 22% [29]. Bonifaz and associates from Mexico reported on 482 patients, mainly actinomycetoma (92%), and the extrapedal accounted for 19.6% [17]. From Senegal, 193 mycetoma patients were reported, eumycetoma was more frequent, and extrapedal mycetoma accounted for 6.7% [31]. The high incidence of extrapedal cases in the Mexican series remains ambiguous. However, in some of these studies, the definition of extrapedal mycetoma was different as it included any site except the foot. Yet, the habits of the affected individuals could offer a partial explanation, given that a majority were engaged in farming. Additionally, these reports indicate a higher prevalence of extrapedal mycetoma among actinomycetoma patients; the cause remains uncertain. However, the prevalence of actinomycetoma in that region may contribute to this pattern.

The results obtained in this study on the patients' demographic characteristics are comparable to various previous reports [17,28,29,30]. Males were affected most and were four times more likely to develop extrapedal mycetoma. This male predominance is similar to previous reports from Sudan and globally [2,17,29,30]. The causes are unclear; however, some hormonal and genetic factors and outdoor activities may justify this male predominance [4,32–34].

Extrapedal mycetoma was more prevalent among young adults, consistent with many reports in the literature [1,2]. The majority of affected patients were in the age group 20–39 years, predominantly students and farmers who constitute a substantial portion of the labour force and serve as the economic backbone. This situation imposes significant adverse socioeconomic consequences on these individuals, their families, and the communities in mycetoma endemic regions. The high incidence of infection among teenagers and young adults can be linked to their engagement in farming activities and frequent barefoot playing, exposing them to agents that cause mycetoma in these endemic areas [1,2,4,17,29].

Family history of mycetoma was recorded only in 11% of the reported patients, which is similar to previous studies [1,2]. Hence, it can be extrapolated that in mycetoma, many predisposing factors can contribute to its causation, such as local household environmental and personal behavioural factors and genetic and immunological factors [4,34].

The duration of mycetoma was less than one year in only 18.5% of patients; consequently, most patients presented with massive lesions with multiple sinuses discharging grains at presentation. The buttock, perineum, head, and back were reported most, and they are in contact with the soil when sitting or lying down. This may validate the hypothesis that mycetoma occurs due to traumatic inoculation of causative organisms [1,2,4,17]. That should be taken into consideration when designing preventive and control programmes.

Regional lymphadenopathy in mycetoma is not an unusual finding that usually occurs due to secondary infection, local immune-complex reactions, or true mycetoma lymphatic spread [35]. In this study, the lymphadenopathy was more frequent with gluteal, buttocks, and inguinal eumycetoma, and direct lymphatic spread is the explanation. Similarly, this study showed axillary lymphadenopathy in a few patients with chest wall mycetoma, in line with Bonifaz's report on two patients with a lymphatic spread from the back to the axillary region [17]. This

study recorded lung spread from gluteal mycetoma, which was probably a blood-borne spread [36].

Although most of the reported patients in this series had massive lesions, proximal varicose veins were rare (0.5%). In comparison, they were reported in (6%) of patients with lower limb mycetoma [37]. It is thought that dilated veins proximal to the mycetoma lesions result from the swift venous return associated with the increased arterial blood flow to the chronic inflammatory granuloma mycetoma lesion. In this series, venous varicosity in the inguinal and suprapubic area was reported in a patient with posterior abdominal wall actinomycetoma, resulting in complete iliac and femoral venous system blockage.

Although mycetoma usually presents with a mass and multiple sinuses discharging grains, abdominal wall mycetoma has a different clinical presentation. In this series, three patients with *Streptomyces somaliensis* caused actinomycetoma presented clinically as abdominal, renal, and retroperitoneal masses and a desmoid tumour. The diagnosis was surgical and histopathological surprises.

One of the important issues in managing mycetoma patients is the high dropout rate among patients. Follow-up dropout among extrapedal mycetoma patients (57%) is slightly higher than in pedal mycetoma patients [38]. The reasons are multifactorial. Most importantly, patients had massive lesions and needed more prolonged treatment, and surgical excisions at these extrapedal sites were difficult and incomplete, hence the high recurrence rate and, thus, the patient's dissatisfaction and dropout. Additionally, most of these patients are of low socio-economic and health education levels and live in remote rural areas without transportation, and it was difficult for them to attend the follow-up. Patients with perineal, inguinal, gluteal, and genital mycetoma are always reluctant to present for medical advice, treatment and follow-up due to shyness and social stigma.

Surgical recurrence in mycetoma is typical, leading to significant morbidity and disability. This study revealed a higher recurrence rate in extrapedal patients (40%) than that reported for pedal mycetoma (27–32%) [1,2]. This may be due to the inadequate surgical excision in this group of patients. In mycetoma, a bloodless field using a tourniquet is vital for adequate excision with a clear margin without disturbing the mycetoma granuloma capsule [39]. This is not applicable in extrapedal mycetoma; hence, there is a higher risk of surgical recurrence. Furthermore, the disease's biological characteristics, the host tissue reactions, and the patients' later presentation due to socio-economic and health education factors can contribute to this high recurrence rate [1,2].

In general, recovery from mycetoma is difficult, especially for eumycetoma patients. The cure rate amongst the study population was markedly low. The eumycetoma group of patients in this study had a relatively low cure rate comparable to other studies [1,2,38].

This study has some limitations; first, it is a descriptive retrospective, and the missing data on the follow-up and dropout among the patients make the evaluation of disease outcomes, especially recurrence and mortality, inaccurate.

In conclusion, this study highlighted the characteristics and outcomes of extrapedal mycetoma patients managed at MRC. There is no clear risk factor for extrapedal mycetoma from the data obtained in this study. The management outcome was unsatisfactory, with a low cure rate, high recurrence, and follow-up dropout rates. Mycetoma endemic areas need systematic and objective health education programmes to encourage early reporting and treatment and, consequently, lessen the medical and socio-economic impacts of mycetoma on patients, families, communities, and the health system at large [40]. To improve treatment outcomes, more efficient and safer drugs are needed to treat eumycetoma, as the current treatment needs more extended time. Patients with mycetoma in the head, ear, eye, oral cavity, face and perineum may need specialised surgical intervention and follow-up [41]. Hence, liaison between the

treating clinician and the speciality of concern is essential to provide the best possible treatment and to track patients during management and follow-up to assess the outcome better.

## Supporting information

**S1 Data. Patients characteristics raw data.**
(XLS)

## Author Contributions

**Conceptualization:** Abubakr Abdalla Mohammed Alhaj, Eiman Siddig Ahmed, Abeer Hassan, Ahmed Hassan Fahal.

**Data curation:** Abubakr Abdalla Mohammed Alhaj, Eiman Siddig Ahmed, Abeer Hassan, Ahmed Hassan Fahal.

**Formal analysis:** Abubakr Abdalla Mohammed Alhaj, Eiman Siddig Ahmed, Abeer Hassan, Ahmed Hassan Fahal.

**Investigation:** Ahmed Hassan Fahal.

**Methodology:** Abubakr Abdalla Mohammed Alhaj, Ahmed Hassan Fahal.

**Supervision:** Ahmed Hassan Fahal.

**Validation:** Abubakr Abdalla Mohammed Alhaj, Eiman Siddig Ahmed, Ahmed Hassan Fahal.

**Visualization:** Abubakr Abdalla Mohammed Alhaj, Eiman Siddig Ahmed, Abeer Hassan, Ahmed Hassan Fahal.

**Writing – original draft:** Abubakr Abdalla Mohammed Alhaj, Eiman Siddig Ahmed, Abeer Hassan, Ahmed Hassan Fahal.

**Writing – review & editing:** Abubakr Abdalla Mohammed Alhaj, Eiman Siddig Ahmed, Abeer Hassan, Ahmed Hassan Fahal.

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
