## [Decision Letter · Decision Letter 0]

16 Feb 2024

Dear Prof. Fahal,

Thank you very much for submitting your manuscript "Epidemiological Observations and Management Challenges in Extrapedal Mycetoma: A Three-Decade Review of 420 Cases" for consideration at PLOS Neglected Tropical Diseases. As with all papers reviewed by the journal, your manuscript was reviewed by members of the editorial board and by several independent reviewers. The reviewers appreciated the attention to an important topic. Based on the reviews, we are likely to accept this manuscript for publication, providing that you modify the manuscript according to the review recommendations. 

Sincerely,

Pascal Del Giudice, MD

Guest Editor

Joshua Nosanchuk

Section Editor

Reviewer's Responses to Questions

**Key Review Criteria Required for Acceptance?**

**Methods**

-Are the objectives of the study clearly articulated with a clear testable hypothesis stated?

-Is the study design appropriate to address the stated objectives?

-Is the population clearly described and appropriate for the hypothesis being tested?

-Is the sample size sufficient to ensure adequate power to address the hypothesis being tested?

-Were correct statistical analysis used to support conclusions?

-Are there concerns about ethical or regulatory requirements being met?

Reviewer #1: For the methods of my answers are yes

Reviewer #2: In the study entitled “Epidemiological Observations and Management Challenges in Extrapedal Mycetoma:

A Three-Decade Review of 420 Cases”, authors present their clinical observations in 420 extra pedal mycetoma patients that attended MRC between January 1991 and December 2021. This descriptive clinical study comprehensively described the clinical characteristics of both Eumycetoma and Actinomycetoma cases and compared the treatment outcomes (Clinical improvements) with each of these character istics. A study like this strongly warrants a publication. 

Some minor comments:

Page 7 -Line 141 : The statistical methods need to be elaborated for the reader to understand the data analysis in detail. 

Page 11 - Lines 215 - 217: For those patients where family history is evident, it will be interesting to see if the type of mycetoma is the same between case and the familial contact.

Reviewer #3: Are the objectives of the study clearly articulated with a clear testable hypothesis stated? Yes

-Is the study design appropriate to address the stated objectives? Yes

-Is the population clearly described and appropriate for the hypothesis being tested? Yes 

-Is the sample size sufficient to ensure adequate power to address the hypothesis being tested? Yes

-Were correct statistical analysis used to support conclusions? Yes

-Are there concerns about ethical or regulatory requirements being met? Not really

**Results**

-Does the analysis presented match the analysis plan?

-Are the results clearly and completely presented?

-Are the figures (Tables, Images) of sufficient quality for clarity?

Reviewer #1: for the results all my answers are yes

Reviewer #2: -Does the analysis presented match the analysis plan?

Yes

-Are the results clearly and completely presented?

Yes

-Are the figures (Tables, Images) of sufficient quality for clarity?

Yes

Reviewer #3: yes

**Conclusions**

-Are the conclusions supported by the data presented?

-Are the limitations of analysis clearly described?

-Do the authors discuss how these data can be helpful to advance our understanding of the topic under study?

-Is public health relevance addressed?

Reviewer #1: Yes the conclusions are supported by the data presented. the limitations are clearly described ie the lack of regular follow-up for a majority of patient ts but this is always observed with mycetoma pateints living for from main hospitals.

The authors discuss these data and th main recommandations are early detection and treatment of mycetoma, especially extra-pedal mycetoma, the necessity of new drugs for eumycetoma and the neccessity of specialised surgery for some extra-pedal localisatioons

Reviewer #2: -Are the conclusions supported by the data presented?

Yes

-Are the limitations of analysis clearly described?

Yes

-Do the authors discuss how these data can be helpful to advance our understanding of the topic under study?

Yes

-Is public health relevance addressed?

Yes

Reviewer #3: yes

**Editorial and Data Presentation Modifications?**

Reviewer #1: This series need minor modifications, the mycetoma research center of Khartoum, is a reference center and present always

main data, results and recommandation about mycetoma

Reviewer #2: Accept

Reviewer #3: Yes I recommended. minor revision

**Summary and General Comments**

Reviewer #1: This is a series of 420 extrapedal mycetoma presented by the mycetoma research center of Khartoum. It is the most important series ever presented on extrapedal mycetoma, few data exists about this aspect of mycetoma. But the main question is: what these authors call an extrpedal mycetoma ? They have to give a definition in the text. For Senegalese authors (ref 31) extrapedal mycetoma is every mycetoma localized outside the foot including upper and lower limps. It explains the difference of percentage of extrapedal localizations between Senegal (39%) and Sudan (4,3%). In Mexico (ref 17), extrapedal mycetoma represent 35,75% of cases, percentage quite similar to the Senegalese results.

Osseous localizations ( 20 periostal reaction, 13 bone destructions) were found in 33 of the cases in the presented series. Would it be possible to have more details about these 33 patients, especially the exact localization. It is said that osseous involvement seems to be rare outside the foot and the hands.

line 301: FNAC is a very interesting method diagnosis...but it is impossible to identify Madurella mycetomatis as only PCR can identify species of Madurella...you can only answer Madurella sp...same remark for histopathology of grains.

line 305: two rare organims were identified: Scedosporium boydi and Aspergillus sp...by culture ?

Reviewer #2: In the study entitled “Epidemiological Observations and Management Challenges in Extrapedal Mycetoma:

A Three-Decade Review of 420 Cases”, authors present their clinical observations in 420 extra pedal mycetoma patients that attended MRC between January 1991 and December 2021. This descriptive clinical study comprehensively described the clinical characteristics of both Eumycetoma and Actinomycetoma cases and compared the treatment outcomes (Clinical improvements) with each of these character istics. A study like this strongly warrants a publication. 

Some minor comments:

Page 7 -Line 141 : The statistical methods need to be elaborated for the reader to understand the data analysis in detail. 

Page 11 - Lines 215 - 217: For those patients where family history is evident, it will be interesting to see if the type of mycetoma is the same between case and the familial contact.

Reviewer #3: document attached.

PLOS authors have the option to publish the peer review history of their article (what does this mean?). If published, this will include your full peer review and any attached files.

Reviewer #1: No

Reviewer #2: Yes: Dr Sundeep Chaitanya Vedithi

Reviewer #3: Yes: Afia Zafar

Figure Files:

Data Requirements:

Reproducibility:

References

---

## [Editor Report · Decision Letter 1]

10 Apr 2024

Dear Prof. Fahal,

We are pleased to inform you that your manuscript 'Epidemiological Observations and Management Challenges in Extrapedal Mycetoma: A Three-Decade Review of 420 Cases' has been provisionally accepted for publication in PLOS Neglected Tropical Diseases.

Best regards,

Pascal Del Giudice, MD

Guest Editor

Joshua Nosanchuk

Section Editor

<style type="text/css">p.p1 {margin: 0.0px 0.0px 0.0px 0.0px; line-height: 16.0px; font: 14.0px Arial; color: #323333; -webkit-text-stroke: #323333}span.s1 {font-kerning: none

</style>

---

## [Editor Report · Acceptance letter]

30 Apr 2024

Dear Prof. Fahal,

We are delighted to inform you that your manuscript, "Epidemiological Observations and Management Challenges in Extrapedal Mycetoma: A Three-Decade Review of 420 Cases," has been formally accepted for publication in PLOS Neglected Tropical Diseases.

Best regards,

Shaden Kamhawi

co-Editor-in-Chief

Paul Brindley

co-Editor-in-Chief
